# A new high-voltage calcium intercalation host for ultra-stable and high-power calcium rechargeable batteries

Zheng-Long Xu [1,2], Jooha Park[1], Jian Wang [3], Hyunseok Moon[4], Gabin Yoon[1], Jongwoo Lim [3], Yoon-Joo Ko[5], Sung-Pyo Cho[5], Sang-Young Lee [4] & Kisuk Kang [1,6,7,8 ✉]

Rechargeable calcium batteries have attracted increasing attention as promising multivalent ion battery systems due to the high abundance of calcium. However, the development has been hampered by the lack of suitable cathodes to accommodate the large and divalent $Ca^{2+}$ ions at a high redox potential with sufficiently fast ionic conduction. Herein, we report a new intercalation host which presents 500 cycles with a capacity retention of 90% and a remarkable power capability at ~3.2 V (vs. $Ca/Ca^{2+}$) in a calcium battery. The cathode material derived from $Na_{0.5}VPO_{4.8}F_{0.7}$ is demonstrated to reversibly accommodate a large amount of $Ca^{2+}$ ions, forming a series of $Ca_xNa_{0.5}VPO_{4.8}F_{0.7}$ ($0 < x < 0.5$) phases without any noticeable structural degradation. The robust framework enables one of the smallest volume changes (1.4%) and the lowest diffusion barriers for $Ca^{2+}$ among the cathodes reported to date, offering the basis for the outstanding cycle life and power capability.

[1] Department of Materials Science and Engineering, Research Institute of Advanced Materials (RAIM), Seoul National University, Seoul, Republic of Korea. [2] Department of Industrial and Systems Engineering, The Hong Kong Polytechnic University, Hung Hom, Hong Kong SAR, China. [3] Department of Chemistry, Seoul National University, Seoul, Republic of Korea. [4] Department of Chemical and Biomolecular Engineering, Yonsei University, Seoul, Republic of Korea. [5] National Center for Inter-University Research Facilities, Seoul National University, Seoul, Republic of Korea. [6] Center for Nanoparticle Research, Institute for Basic Science (IBS), Seoul National University, Seoul, Republic of Korea. [7] Institute of Engineering Research, College of Engineering, Seoul National University, Seoul, Republic of Korea. [8] School of Chemical Bioengineering, College of Engineering, Seoul National University, Seoul, Republic of Korea. ✉email: matlgen1@snu.ac.kr

The growing demands for electric vehicles and stationary energy storage systems have motivated exhaustive efforts to explore new types of batteries with a higher energy density, longer life, and lower cost than that offered by the current lithium ion batteries (LIBs)[1–3]. Multivalent ion batteries have been considered as one of the alternative solutions, because multivalent ion insertion/extraction is associated with doubled or tripled electron transfer per ion in the intercalation reaction, potentially leading to higher specific energy densities than that the monovalent ions can carry[4]. Recently, rechargeable batteries based on magnesium ion[5], aluminum ion[6], zinc[7], and calcium ion have thus received much attention and obtained discernible progress in battery performance. Among the multivalent battery systems, calcium ion batteries (CIBs) are capable of offering the highest voltage due to the low reduction potential of Ca/Ca$^{2+}$ with −2.9 V (vs. standard hydrogen potential, SHE), which is lower than −2.4 V (vs. SHE) for magnesium, −1.7 V (vs. SHE) for aluminum, and −0.76 V (vs. SHE) for zinc[8,9], enabling a high energy density with the voltage comparable to that of LIBs. In addition, calcium ion presents the smallest polarization strength of 10.4 (vs. 14.7 for magnesium ion, 16 for zinc ion, and 24 for aluminum ion)[10], suggesting comparatively fast mobility of divalent calcium ions in hypothetical insertion host materials. Together with the abundance of calcium in earth crust and its environmental compatibility, these properties appeal CIBs as the next generation post-LIBs.

Recent efforts on the development of negative electrodes could bring the rechargeable calcium chemistries a step closer to a practically feasible battery system[11]. While a reversible stripping/plating of calcium metal electrode had been regarded extremely challenging due to the formation of ion-blocking layer on the surface of calcium metal, several research groups successfully verified that it can be resolved by employing new electrolyte systems such as Ca(BF$_4$)$_2$ in ethylene carbonate/propylene carbonate (EC/PC)[12], Ca(BH$_4$)$_2$ in tetrahydrofuran (THF)[13], and Ca[B(hfip)$_4$]$_2$ in dimethoxy ethane (DME), demonstrating elemental calcium metal as an important CIB anode[14]. Moreover, it was recently reported that a large amount of Ca$^{2+}$ can be reversibly intercalated to a graphite anode through a co-intercalation reaction over 200 cycles, presenting its promise as a stable anode for CIBs[15]. In coping with the remarkable advancement of anodes for CIBs, efforts have been also placed on the discovery of cathode over the past years[11]. Various materials groups have been proposed such as layered materials (i.e., TiS$_2$[16], V$_2$O$_5$[17], α-MoO$_3$[18]), Prussian blue analogues (i.e., MnFe(CN)$_6$[19]) and transition metal oxides (i.e., Ca$_x$Mn$_2$O$_4$[20]), which could exhibit the capability to store calcium ions and the promise for the use as cathode. Nevertheless, the cyclic performance of these proposed cathodes seldom exceeded 100 cycles, and few cathodes could deliver a reasonably high capacity at practically important current rates, incompatible with the advanced CIB anodes[16–20]. It is likely due to the relatively large ionic radius and divalent nature of Ca$^{2+}$ compared to monovalent ions (i.e., Li$^+$ and Na$^+$), which make the intercalation kinetics generally sluggish in diffusion channels of intercalation hosts. Moreover, a large calcium ion intercalation in the host is supposed to cause an extended volume change of the host, which triggers a premature degradation of the cathode structure[11]. The discovery of a reliable calcium cathode material or strategy that can mitigate these issues would expedite the development of the CIBs, and thus has long been awaited.

In our study exploring a new CIB cathode here, we attempted to search for a material chemistry with a rigid open framework that is less sensitive to the volume change arising from the large guest ion insertion/extraction. In this regard, a success-proven polyanion-based cathode for sodium ion battery, Na$_{1.5}$VPO$_{4.8}$F$_{0.7}$ (NVPF)[21–23], attracted our immediate attention. According to our previous studies, NVPF has shown interesting properties as a host material for sodium ions, exhibiting one of the lowest volume changes with (de-)sodiation (~2.9%) and fast sodium ion diffusion with low activation barriers (30–330 meV) owing to its unique open framework, thus delivering excellent cycle and power performances[22]. Moreover, the high redox potential (3.9 V vs. Na/Na$^+$) and the latent multi-redox capability of vanadium ion (V$^{3+}$/V$^{4+}$/V$^{5+}$) could contribute to a high energy density in the sodium system, which are also supposed to be beneficial when applied to CIB cathode. Herein, exploiting the structurally robust and open-framework NVPF host, we show that the desodiated NVPF can function as a stable and fast-kinetic calcium ion intercalation host and thus be a strong contender to the currently available cathodes for rechargeable CIBs. It is demonstrated that the NVPF-based host allows reversible Ca$^{2+}$ ion intercalation and deintercalation at ~3.2 V (vs. Ca/Ca$^{2+}$) in calcium cells with the capacity fading rate of 0.02% per cycle over 500 cycles, which records one of the lowest values reported to date for CIB electrodes. Equally important is that its power capability outperforms most of the existing cathodes for CIBs. The origin of the outstanding performance is elucidated through comprehensive experimental and theoretical investigations.

## Results

**Electrochemical properties of NVPF cathode in CIBs.** In order to verify the intercalation capability of Ca$^{2+}$ ion into NVPF framework, we constructed an electrochemical cell as illustrated in Fig. 1a, which consists of desodiated NVPF as the working electrode, activated carbon as a counter electrode, and an electrolyte of 1 M Ca(PF$_6$)$_2$ in EC/PC[19]. Prior to electrochemical tests in the calcium cell, the desodiated NVPF electrode was prepared by charging the NVPF electrode in a separate sodium electrochemical cell to 4.5 V vs. Na/Na$^+$, resulting in []$_{1.0}$Na$_{0.5}$VPO$_{4.8}$F$_{0.7}$ ([]: vacancy) compound (see Supplementary Fig. 1 for details), according to our previous work[22]. The activated carbon was used as the counter electrode to prevent the complexity arising from unwanted side reactions caused by the use of calcium metal, and is known to be electrochemically active in calcium electrolytes via simple (de)adsorption reaction[19]. The galvanostatic discharging/charging of the calcium cell was first performed at 25 mA g$^{-1}$ between −1.0 and 1.5 V vs. activated carbon electrode, which corresponds to the voltage range of 1.75 and 4.25 V vs. Ca/Ca$^{2+}$, as presented in Fig. 1b. (The voltage calibration with respect to the value vs. Ca/Ca$^{2+}$ is described in the experimental section and Supplementary Figs. 2, 3[24].) Interestingly, the desodiated NVPF electrode could deliver an appreciable discharge capacity in the calcium electrochemical cell, implying a significant amount of calcium stored in the desodiated NVPF structure. The initial discharge capacity was ~78 mAh g$^{-1}$, which immediately increased to 88 mAh g$^{-1}$ in the following discharge cycles. A negligible change in the capacity or the electrochemical profile was observed during the subsequent ten cycles (Fig. 1b), indicating excellent reversibility for Ca$^{2+}$ insertion and extraction in desodiated NVPF cathodes. As depicted in Supplementary Fig. 3, the electrochemical profile remained almost unchanged even after 50 cycles, delivering a reversible capacity of 87 mAh g$^{-1}$ at 25 mA g$^{-1}$. The calcium intercalation into the NVPF host could be directly confirmed from the chemical analysis of the fully discharged electrode. As tabulated in Supplementary Table 1, the electrode composition was determined to be Ca$_{0.38}$Na$_{0.56}$VPO$_{4.8}$F$_{0.7}$ after the full discharge in the cell from the inductively coupled plasma mass spectrometry (ICP-MS) measurements. It is markedly consistent with the estimated composition of the fully calciated NVPF, i.e., Ca$_{0.35}$Na$_{0.5}$VPO$_{4.8}$F$_{0.7}$, based on the capacity delivered in Fig. 1b assuming the Ca$^{2+}$ ion intercalation. After

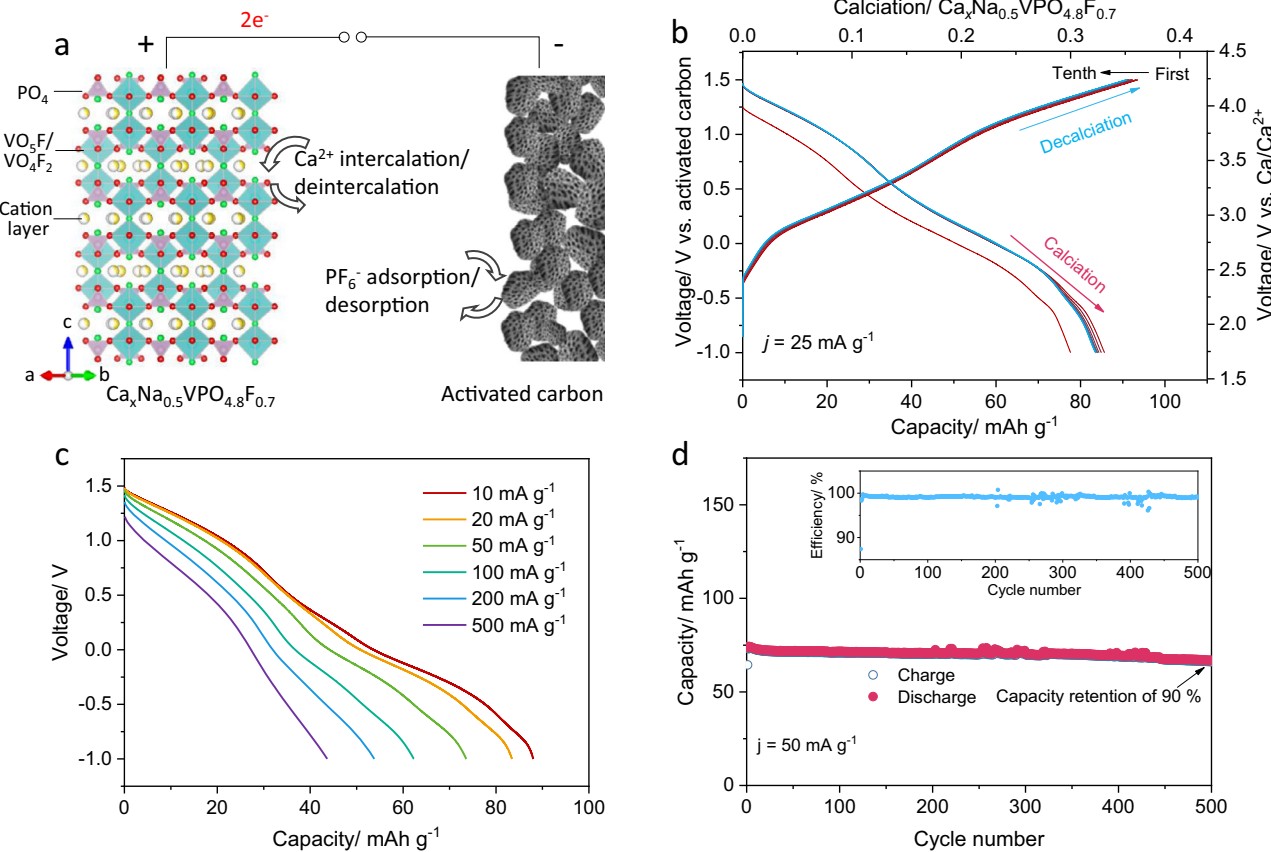

**Fig. 1 Electrochemical performance of desodiated NVPF cathode in CIBs. a** Schematic illustration of the working principle of a CIB where the insertion/ removal of $Ca^{2+}$ ions in $Ca_xNa_{0.5}VPO_{4.8}F_{0.7}$ cathode is accompanied with adsorption/desorption of $PF_6^-$ on activated carbon anode. The $VO_5F/VO_4F_2$ octahedra, $PO_4$ tetrahedra and cation (Na, Ca, and vacancy) units are shown in **a**. **b** First ten cycles of discharge/charge profiles at a current density of $j =$ 25 mA g$^{-1}$ between −1 and 1.5 V. **c** Calcium insertion capacities at increasing current densities from 10 to 500 mA g$^{-1}$. **d** Long-term discharge/charge capacities and corresponding Coulombic efficiencies at 50 mA g$^{-1}$ for 500 cycles.

recharging of the electrode, the composition was measured to be $Ca_{0.05}Na_{0.51}VPO_{4.8}F_{0.7}$, confirming that the capacity in the calcium cell was originated from the reversible calcium intercalation/deintercalation.

The rate capability of the new calcium cathode was further investigated by probing the discharging responses at increasing current densities, as shown in Fig. 1c. When the current density increased to 100 mA g$^{-1}$ corresponding to ~1 C rate, the discharge capacity only slightly reduced from 87 to 62 mAh g$^{-1}$. Even with the 50-folds increase in the current density to 500 mA g$^{-1}$, nearly 50% of the specific capacity could be still retained, indicating a respectable rate capability. While this power performance does not reach that of the conventional LIBs based on the monovalent lithium ion conduction, it outperforms most other intercalation compounds reported in previous CIBs, e.g., 38 mAh g$^{-1}$ at 400 mA g$^{-1}$ for $VOPO_4·2H_2O$ cathode[25], and 45 mAh g$^{-1}$ at 117 mA g$^{-1}$ for $NaV_2(PO_4)_3$ cathode[26]. Moreover, it is noteworthy that the primary particles of our material are a few micrometers in size (Supplementary Fig. 1a), which indicates a further promise in the high power capability by nanostructure engineering. Detailed analysis on the kinetics of the calcium intercalation in NVPF structure will be discussed later in the following section. Figure 1d presents the stability of the NVPF cathode in a calcium electrochemical cell during the extended cycles at a current density of 50 mA g$^{-1}$. It illustrates that the initial discharge capacity of 75 mAh g$^{-1}$ is well retained, and the capacity retention is over 90% after 500 cycles, rendering a remarkably low capacity fading rate of 0.02% per cycle. To

estimate the standing of current desodiated NVPF cathode among its peers, we summarize the electrochemical performance of the state of the art cathodes in CIBs as listed in Supplementary Table 2[10,18,19,25,27–29]. It clearly illustrates that our NVPF cathode delivers the highest cyclic stability and power capability at a high redox voltage (3.2 V in this work in comparison with 2.4 V for $Mg_{0.25}V_2O_5·H_2O$[10], 1.3 V for α-$MoO_3$[18], and 2.8 V for $VOPO_4·H_2O$[25]) for CIBs.

**Calcium storage mechanism of NVPF cathode.** In order to elucidate the remarkable performance of the NVPF electrode in calcium electrochemical system, we carefully investigated its charge/ discharge mechanism. First, the structural evolution of the electrode was probed in real time by synchrotron in situ XRD measurements. The XRD patterns were periodically collected every 4 min when the cell was galvanostatically cycled at 15 mA g$^{-1}$. Figure 2a depicts an augmented view of the main XRD peaks of the NVPF electrode during the discharge and charge reactions. (See Supplementary Fig. 4 for the full diffractograms.) It clearly indicates that the electrochemical calciation/decalciation is accompanied by the gradual structural transformation of the NVPF, which is highly reversible, a testament to the excellent reversibility of the CIB cell. A closer examination of the XRD patterns reveals that a continuous shift of peaks occurs during the first half of the calciation, suggesting the solid-solution reaction with the calcium intercalation. On the other hand, during the second half of the calciation into NVPF, a hint of the two-phase coexistence was detected in the region of 28.1–28.4°. As marked with red dash lines in Fig. 2b, the peak at 28.4° gradually

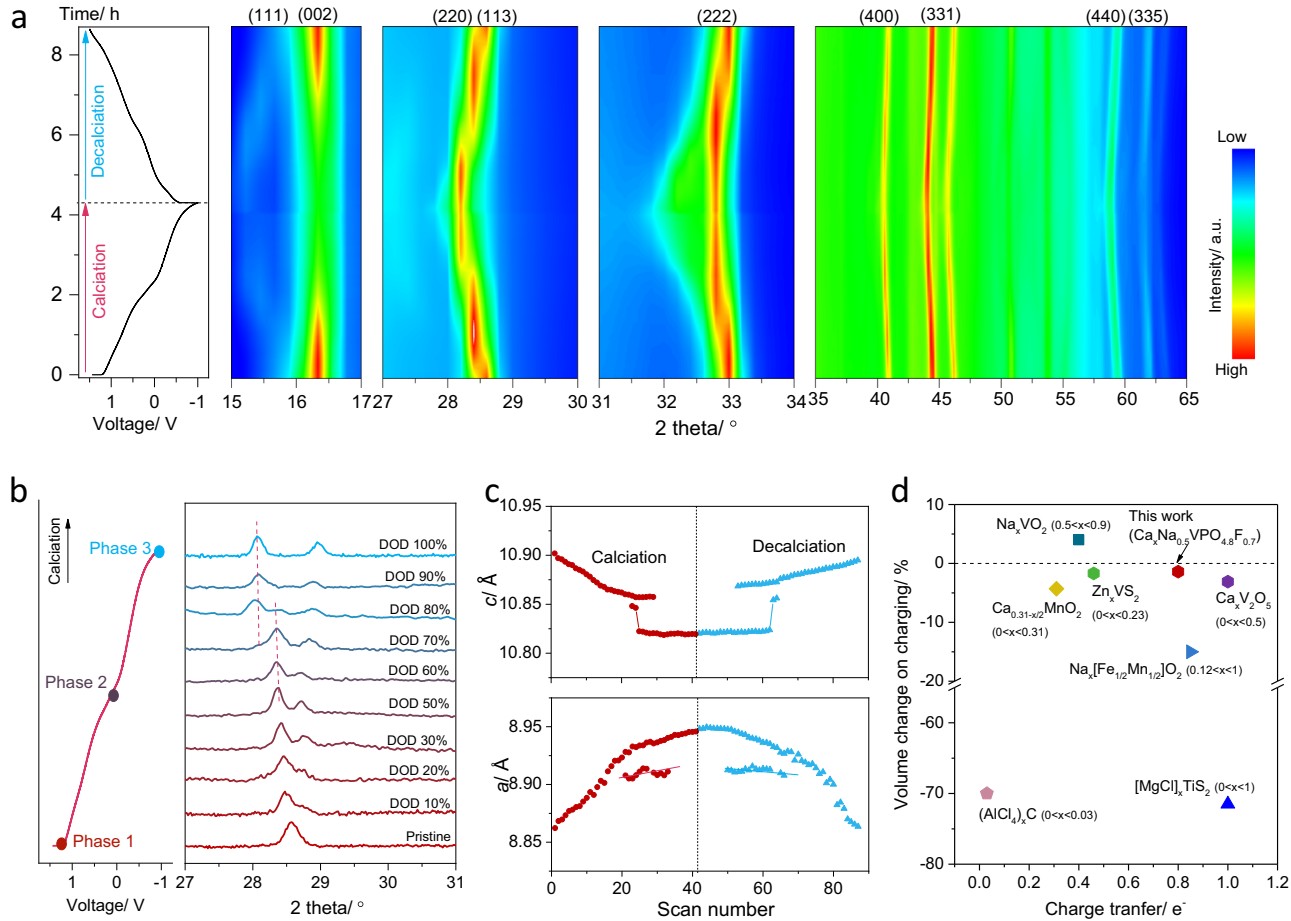

**Fig. 2 Synchrotron in situ XRD analysis of the structural evolution of desodiated NVPF cathode during calciation and decalciation. a** Discharge/charge voltage profile and the corresponding in situ XRD intensity map, where blue refers to low intensity and red corresponds to high intensity. **b** Evolutions of (220) and (113) reflections of the in situ XRD patterns at different depth of discharge; the left shows the typical voltage profile of desodiated NVPF upon calciation; Phase 1 and 3 represent the pristine phase (desodiated NVPF) and the fully calciated phase, respectively. **c** Changes in lattice parameters of *a* and *c* during calciation and decalciation derived from in situ XRD patterns in **a**. **d** Comparison of volume changes on charging in various cathodes as a function of charge transfer number[6,17,28,30–33].

disappears, while that at 28.1° emerges from DOD (depth of discharge) 50% to DOD 100%. It implies that a local segregation of two Ca configurations has arisen in the structure (i.e., the occurrence of biphasic domains). This behavior is reminiscent of the Na insertion process in NVPF, where a stable intermediate phase at $x = 1$ in $Na_xVPO_{4.8}F_{0.7}$ induced the biphasic reaction at a high Na content region[22]. Analogically, a stable intermediate phase for calciated NVPF or/and short-range ordering of Ca in basic units may be attributable to the two-phase region. More studies are necessary in the future to reveal the origin of the two-phase reaction and to elucidate the detailed structure of the intermediate calcium phase.

In Fig. 2c, lattice parameters *a* and *c* are plotted as a function of calciation and decalciation, as determined by the Rietveld refinement of the XRD profiles. It shows that the lattice parameter *c*, which corresponds to the interlayer distance of the NVPF crystal structure in Fig. 1a, decreases with the calciation, whereas the lattice parameter *a* corresponding to the basal dimension increases. The reduction of the lattice parameter *c* is attributed to the decreased electrostatic repulsion between $O^{2-}$ layers, as the $Ca^{2+}$ layers that screen the electrostatic repulsion between $O^{2-}$ become continuously filled. On the other hand, the increase in the lattice parameter *a* is attributable to the expansion of vanadium octahedra with the electrochemical reduction of vanadium ions. Upon decalciation, the reverse behavior was

observed for the lattice parameters of *a* and *c*, manifesting the reversibility of the NVPF structure during calciation and decalciation. Noteworthy is that the variation of *c* and *a* lattice parameter is 0.8% by contraction and 1.1% by expansion, respectively, with the full calciation, which leads to an ultra-small volume variation of 1.4%. This value is unusually small considering the large calcium ion intercalation to the crystal, and is even smaller than those of most intercalation compounds for post LIB chemistry. In Fig. 2d, we comparatively displayed the volume expansion of various cathodes for CIB (and for several post LIB systems such as Al, Mg, Zn, and Na ion batteries) upon charging[6,17,28,30–33]. For fair comparisons, the volume change was plotted with the amount of charge transfer accompanied with the intercalation. It clearly illustrates that the calcium intercalation in NVPF cathode involves the lowest volume change among peers even with a substantial amount of the charge storage. To our surprise, the volume change of NVPF cathode with calcium (1.4%) is only half of that observed for the sodium intercalation in NVPF, which is known to be 2.9%[22]. We suspect that, given the similar ionic size of $Ca^{2+}$ and $Na^+$ (2.0 Å for $Ca^{2+}$ and 2.04 Å for $Na^+$)[10], the divalent $Ca^{2+}$ only needs to occupy half of the interstitial vacancies in the intercalation host compared with the monovalent $Na^+$ to retain the high specific capacity, thus involves a smaller volume change in the host. It proposes an unexpected

merit of the multivalent ion intercalation in batteries, which requires a smaller quantity of guest ion intercalants for a given capacity than the monovalent ion counterpart, leading to a smaller variation in the host structure during the electrochemical reaction. The positive correlation between the small volume variation and the stable cyclic property of electrodes has also been proven in the well-known zero-strain $Li_4Ti_5O_{12}$ anodes in LIBs[34]. We believe that the NVPF framework involving a small volume change with the calcium intercalation has also contributed to the outstanding cyclic stability demonstrated in Fig. 1d.

The detailed structural analysis of the NVPF cathode during calcium insertion/extraction was further carried out using high-resolution powder XRD (HRPD), X-ray absorption near-edge structure (XANES), and $^{13}Na$ nuclear magnetic resonance (NMR) measurements. The Rietveld refinements of the pristine, desodiated, calciated, and decalciated NVPFs were carried out using HRPD, and their results are provided and tabulated in Supplementary Fig. 5 and Supplementary Tables 3–6, respectively. The Na and Ca stoichiometry derived from the refinement matches the electrochemical and ICP-MS results. Moreover, the $a$- and $c$-axis lattice parameters changes as derived from HRPD upon calciation/decalciation are consistent with the in situ XRD results. In Fig. 3a, we comparatively plotted the change in the occupancies of Na1 and Na2 sites in the NVPF, which are known to be the primary sites for the sodium ions in the previous works[21–23]. (Na1 and 2 sites are also schematically shown in Fig. 3e.) The figure indicates that Na1 sites are mainly deintercalated in the desodiation process, which serve as vacant interstitial sites for the following calcium insertion reaction. It was also consistently found that $Ca^{2+}$ ions mainly occupy vacancies on Na1 sites, presenting a similar crystallographic structure with the pristine NVPF. On the other hand, the Na2 sites were observed to be negligibly occupied by $Ca^{2+}$ ions, thus do not noticeably contribute to the overall capacity. This preferred contribution of Na1 sites in the capacity is similar to the sodium intercalation/deintercalation behavior of NVPF in the sodium electrochemical system[35].

The V K-edge XANES characterization in Fig. 3b discloses that discharging/charging processes were accompanied by the reduction and oxidation of the vanadium ion. Pristine NVPF shows a dominant pre-edge peak at 5469.3 eV, indicating the coexistence of $V^{4+}$ and $V^{3+}$, which was also previously confirmed by electron paramagnetic resonance experiment[23]. After the pre-desodiation, the pre-edge peaks shifted to higher energies at 5469.7 and 5471 eV, signifying the oxidation of the vanadium ion[22]. When $Ca^{2+}$ ions were inserted in the desodiated NVPF, the reduction of vanadium ion was identified with a negative shift of the pre-edge peak to that for the pristine NVPF. During the reverse decalciation reaction, the vanadium ion returned to the original oxidation state, implying that the vanadium redox reaction is fully reversible in CIBs. Further probe on the NVPF cathode during discharging/charging was conducted using $^{23}Na$ NMR, as shown in Fig. 3c. The $^{23}Na$ NMR spectrum of the pristine NVPF exhibits two major resonances at 84.5 and 127.6 ppm, which are assigned to the Na sites neighboring $V^{4+}$ and $V^{3+}$ (Fig. 3d), respectively, as the $V^{4+}:V^{3+}$ in the structure is 4:1 ratio[23]. The fully desodiated NVPF presents a low resonance at 23.6 ppm, a sign of high oxidation state of vanadium ion (i.e., $V^{5+}$) with the reduced number of unpaired electrons[35]. When $Ca^{2+}$ ions were inserted in the desodiated NVPF, a major peak was detected at a higher resonance of 96.2 ppm, signifying the reduction of vanadium ion. Considering that Na1 was dominantly extracted for desodiated NVPF from the XRD refinement, we suspect that the broad resonance at 96.2 ppm originates mainly from sodium ions in Na2 sites and some residual Na1 affected by the adjacent $Ca^{2+}$ ions. The $^{23}Na$ resonance peak is more broadened than that of

pristine NVPF, which is attributable to the weaken interaction between residual Na ions and the paramagnetic vanadium ions in the presence of Ca ions[21]. After the calcium extraction, the $^{23}Na$ resonance is still observable and shifts back to 37.8 ppm, close to that for desodiated NVPF. It manifests that a reversible calcium extraction takes place with the remaining structural sodium ions in the NVPF framework, as illustrated in Fig. 3e. The difference between the $^{23}Na$ peak position of decalciated NVPF and desodiated NVPF (23.6 ppm vs. 37.8 ppm) is possibly arising from partial Na ion migration from Na2 sites to Na1 sites (see the Na occupancies in Supplementary Tables 4 and 6).

The calciation of Na-vacancy layers in desodiated NVPF could be directly confirmed by scanning transmission electron microscopy (STEM) and electron energy loss spectroscopy (EELS) measurements for the calciated NVPF electrode. Figure 3f presents the high-angle annular-dark-field (HAADF, left) and annular-bright-field (ABF, right) STEM images viewed along [110] direction. The Z-contrast of HAADF image illustrates that zigzag stackings of $VO_5F$ (or $VO_4F_2$) octahedral/$PO_4$ tetrahedra units (bright dots) form the layered structure, where calcium or sodium ions are nearly invisible (dark) between the layers, as schematically illustrated in Supplementary Fig. 6. On the other hand, ABF image on the right panel along with the EELS analysis clearly indicates the presence of calcium ions in the layer. Figure 3g displays the local composition of calciated NVPF determined from the EELS fine structure of V L-edge, Ca L-edge, and O K-edge spectra. Periodical appearance of calcium ions was evidently observed where the intensity for vanadium was low, suggesting calcium ions are located between $VO_5F$ (or $VO_4F_2$)/$PO_4$ layers. The series of characterization along with the direct observation of calcium approve that the electrochemical reaction in CIB occurs via the reversible $Ca^{2+}$ intercalation at sodium vacant sites in the desodiated NVPF with vanadium redox reactions (Fig. 3e and Supplementary Fig. 6).

**Kinetic behavior of NVPF cathode in CIBs**. Having verified the insertion of calcium ions in the structure, we attempted to unveil the kinetic properties of $Ca^{2+}$ ion migration in the NVPF structure. The diffusion coefficients were measured as a function of the calcium concentration in $Ca_xNa_{0.5}VPO_{4.8}F_{0.7}$ using the galvanostatic intermittent titration technique (GITT) in Supplementary Fig. 7. Figure 4a plots the chemical diffusion coefficients of calcium ions measured both at insertion and extraction processes from the GITT. It reveals that the diffusion coefficients range from $8.9 \times 10^{-12}$ to $4.1 \times 10^{-11}$ cm$^2$ s$^{-1}$, which is superior or comparable to the typical values of multivalent ions in the intercalation hosts such as $10^{-17}$ cm$^2$ s$^{-1}$ for $Mg^{2+}$ diffusion in cation-deficient $TiO_2$[36], theoretically predicted diffusion coefficient of $5 \times 10^{-11}$ cm$^2$ s$^{-1}$ for $Mg^{2+}$-containing spinel oxides[37], $10^{-18}$–$10^{-20}$ cm$^2$ s$^{-1}$ for $Al^{3+}$ diffusion in cubic $Cu_{0.31}Ti_2S_4$[38] and $10^{-17}$–$10^{-15}$ cm$^2$ s$^{-1}$ for $Zn^{2+}$ in $ZnNaV_2(PO_4)_3$[39]. It is also worthy of mentioning that the diffusivity is even comparable to those of lithium ion diffusion in conventional LIB cathode materials such as $LiCoO_2$ ($10^{-11}$–$10^{-13}$ cm$^2$ s$^{-1}$)[40] and $LiFePO_4$ ($10^{-10}$–$10^{-16}$ cm$^2$ s$^{-1}$)[41], highlighting a superior $Ca^{2+}$ ion diffusion kinetics in the current work. A closer look at the trend in Fig. 4a indicates a concentration-dependent calcium ion diffusion coefficient, which presents higher values at the low calcium concentrations in $Ca_xNa_{0.5}VPO_{4.8}F_{0.7}$ ($0 \le x \le 0.175$). This result is consistent with the in situ electrochemical impedance spectroscopy (EIS) analysis that the charge transfer resistance is close to the minimum at the vacancy-rich states of charging or discharging (Supplementary Fig. 8). Note that the feature of concentration-dependent diffusion kinetic was widely observed in lithium ion diffusion in other layered compounds such as

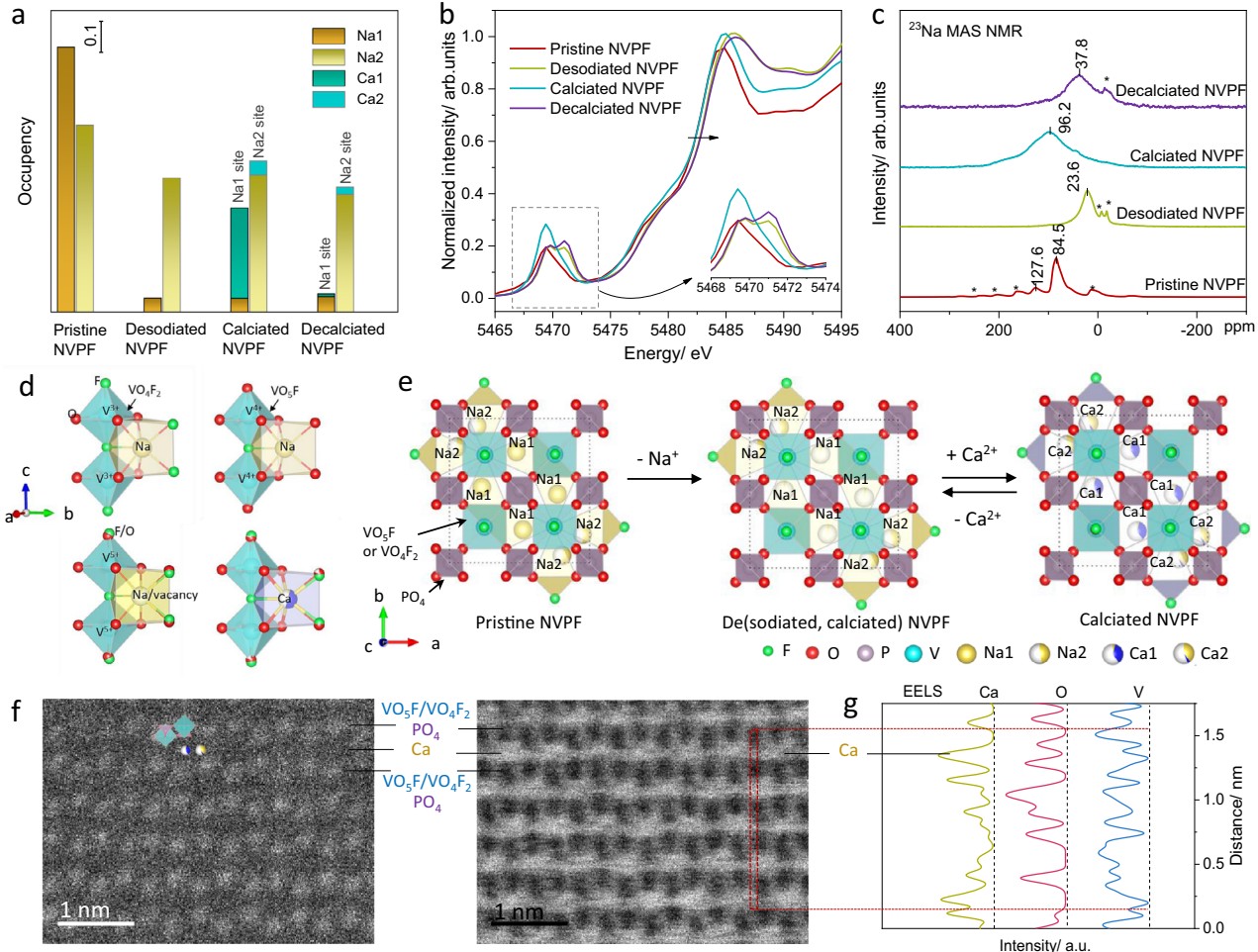

**Fig. 3 Detailed structural and morphological characterization of desodiated NVPF after calciation and decalciation. a** Cation occupancy at Na1 and Na2 sites for pristine, desodiated, calciated, and decalciated NVPFs derived from the high resolution XRD patterns and Rietveld refinement in Supplementary Fig. 5 and Supplementary Table 3–6, respectively. **b** Vanadium K-edge XANES spectra of the pristine, desodiated, calciated, and decalciated NVPFs. The inset shows an enlarged image of the pre-edge region. **c** $^{23}$Na MAS NMR spectra of the pristine, desodiated, calciated, and decalciated NVPFs at a MAS spinning speed of 15 kHz; the asterisks are assigned to spinning side bands. The average valent state of vanadium in pristine NVPF was determined to be +3.8, which corresponds to a 4:1 ratio for $V^{4+}:V^{3+}$, indicating major Na ions are surrounded by $V^{4+}$ units in the pristine NVPF, which agrees with the previous report[23]. The Na extraction occurs with vanadium oxidation, which results in an overall negative shift of the $^{23}$Na resonance peak (~23.6 ppm) due to the reduced number of unpaired electrons on charge[35]. **d** Four types of possible local environments for vanadium bioctahedra in Na ion or Ca ion. **e** Schematic illustration of the crystallographic evolutions of desodiated NVPF for calciation and decalciation reactions. The F, O, P, V, Na, and Ca atoms are in color of green, red, pink, aqua, yellow, and blue, respectively. The white areas in Na and Ca refer to vacancy concentration. The Na1 and Na2 sites are shared by both the calcium and sodium occupancies, where the fraction of blue and yellow in the circle roughly represents the ratio. **f** HAADF-STEM (left) and ABF (right) images for fully calciated NVPF. The $VO_5F$ (or $VO_4F_2$)/$PO_4$ and Ca layers are marked. **g** EELS spectra to show the Ca, O, and V components in red area in **f**.

$Li_xCoO_2$ and $Li_xTiS_2$[42]. Compared with these intercalation compounds, the variation of the diffusivities is comparably small with the states of charge, which is speculated to be correlated with the structural aspect involving minimal volume change of the NVPF.

In order to gain more insight into the diffusion behavior in $Ca_xNa_{0.5}VPO_{4.8}F_{0.7}$, we conducted density functional theory calculations. As previously reported for lithium and sodium analogues[22,23], there are two representative diffusion pathways for the intercalating cation in NVPF structure; one is the intra-unit pathway, and the other is the inter-unit pathway, as illustrated with yellow circle arrows and green arrows in Fig. 4b, respectively. Our calculations revealed that the activation barrier for the $Ca^{2+}$ ion diffusion is as low as ~243 meV through the intra-unit pathway (Fig. 4c), and ~606 meV through the inter-unit pathway (Fig. 4d). Since calcium ions should migrate through both paths for the overall de/intercalation reaction[23], the

inter-unit hopping would serve as the rate-determining step with the overall activation barrier of ~606 meV. Noteworthy is that the activation barrier of ~606 meV is significantly lower than 750 meV for $Ca_xCo_2O_4$[43], ~1.7 eV for $\alpha$-$V_2O_5$[44], and 2 eV for perovskite $Ca_xMoO_3$[45] in CIBs, elucidating the origin of the fast calcium intercalation kinetics in NVPF compared with other reported cathodes[11]. Additionally, the two-dimensional diffusion nature of the NVPF framework is also beneficial for the fast calcium mobility, while the common one-dimensional diffuser such as $LiFePO_4$ would be sensitively affected by the presence of defects[46].

**Full CIBs.** Finally, in order to verify the practical aspect of NVPF cathode in CIBs, a prototype Ca ion full cell, consisting of calcium metal anode and desodiated NVPF cathode, was

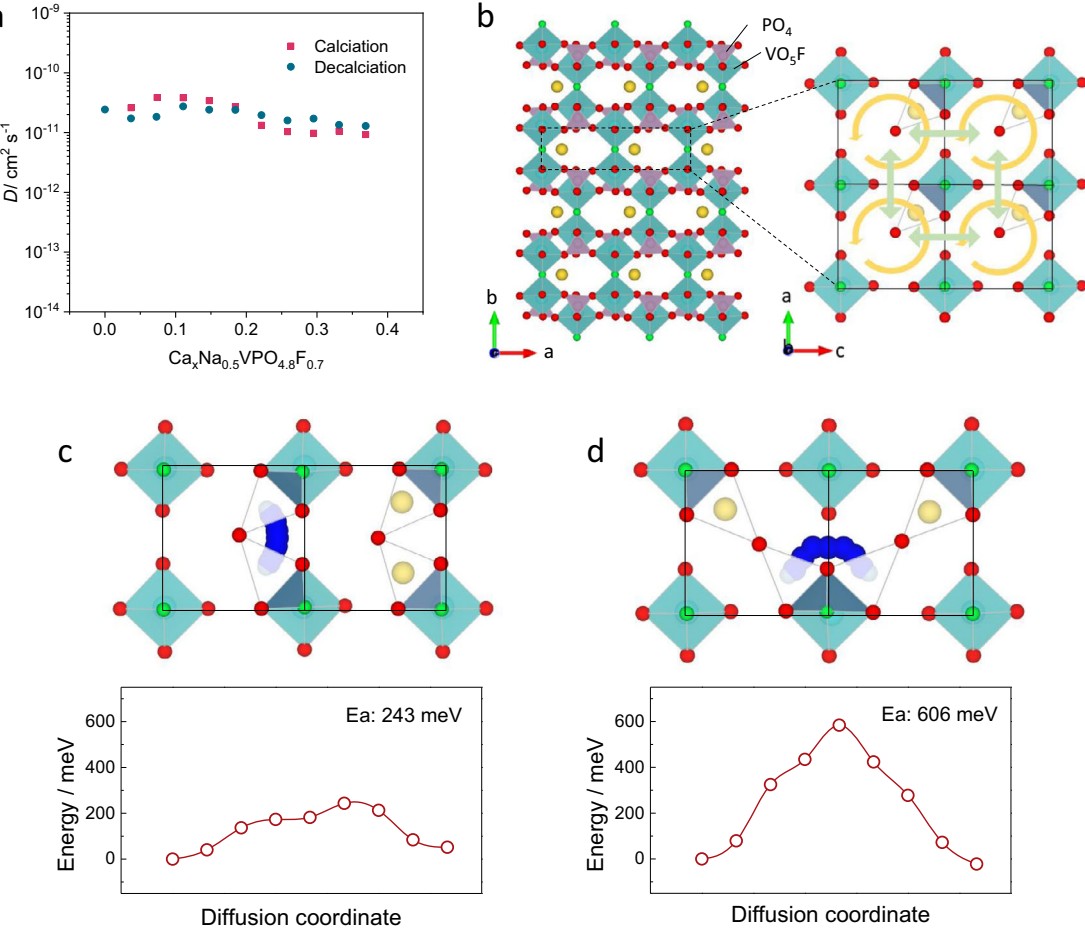

**Fig. 4 Kinetic behavior of the desodiated NVPF in CIBs. a** Evolution of the diffusion coefficient (D) of $Ca^{2+}$ in $Ca_xNa_{0.5}VPO_{4.8}F_{0.7}$ as a functional the Ca concentration derived from GITT. **b** Schematic representing the two Ca diffusion pathways in $Ca_xNa_{0.5}VPO_{4.8}F_{0.7}$. Yellow circle arrows and green arrows represent intra-unit and inter-unit paths, respectively. Ca diffusion pathways and diffusion energies for **c** intra-unit and **d** inter-unit diffusion. Ca, Na atoms, and $VO_5F$ octahedral are in blue, yellow balls, and aqua rhombohedron, respectively.

constructed and cycled at 75 °C in carbonate electrolyte[12]. Supplementary Fig. 9 shows that the cell presents the characteristic discharge/charge profile in consistent with those observed in desodiated NVPF//activated carbon cells in Fig. 1b. The NVPF cathode could deliver a specific capacity of 85 mAh g$^{-1}$ approximately at 3.1 V, yielding an energy density of about 263 Wh kg$^{-1}$ with the calcium metal. While the capacity degrades rapidly due to the passivation of calcium metal in the carbonate-based electrolyte, it confirms that NVPF cathode is capable of functioning as a cathode. Furthermore, the feasibility of the NVPF cathode was explored in combination with other reported anodes by constructing the all-intercalation-based CIBs for the first time. In Supplementary Fig. 10, we assembled the CIBs using desodiated NVPF cathode, calciated graphite anode in a dual phase liquid electrolyte at room temperature. Since the use of graphite anode in CIB requires the specific choice of electrolyte system for the co-intercalation[15], the dual phase electrolyte was designed in regard to immiscibility of two liquid phases[47] and chemical compatibility between active material and electrolyte. An all-intercalation-based full cell showed the reversible capacity retention of 63% over 50 cycles, showing the promise of NVPF cathode in diverse cell configurations. Overall, the remarkable cyclic stability, high power capability, and high energy density for the new cathode propose its feasible applications in the future high energy CIBs.

## Discussion

We have shown that a new intercalation-type cathode unlocks the electrochemical activity towards multivalent calcium ion batteries. The NVPF-based cathode delivered a reversible capacity of 87 mAh g$^{-1}$, an unprecedentedly high capacity retention of 90% over 500 cycles and high rate capabilities, which could rival with state of the art cathodes reported for CIBs thus far. Structural analyses coupled with in situ XRD, solid state NMR, XANES, and STEM revealed the reversible insertion and removal of $Ca^{2+}$ ions in the NVPF framework with an ultra-small volume change, which records one of the smallest values for multivalent ion intercalations. Electrochemical kinetic studies combined with first principles calculations demonstrated the superior diffusion coefficients and low activation barriers for $Ca^{2+}$ ion diffusion, offering insight into the origins of the high-power calcium intercalation cathodes. This work successfully nudges the reversible $Ca^{2+}$ intercalations in a polyanion-based cathode, which provides an unexplored pathway toward the realization of stable and high-power cathodes in CIBs.

## Methods

**Materials synthesis.** The $Na_{1.5}VPO_{4.8}F_{0.7}$ cathode material was synthesized by solid-state reactions[22]. Specifically, $VOPO_4$ powder was first prepared by mixing a stoichiometric amount of $V_2O_5$ (Sigma Aldrich, 95%) and $NH_4H_2PO_4$ (Sigma Aldrich, 99%) by ball milling and annealing at 750 °C for 4 h in Ar flow. Then, a stoichiometric amount of $V_2O_5$ and $NH_4H_2PO_4$ was blended with 20 wt% super

P by ball milling before sending to heat treatment at 850 °C for 2 h in air, to form $VPO_4$. The as-obtained $VOPO_4$, $VPO_4$ powders were later mixed with NaF (Sigma Aldrich, 99%) and $Na_2CO_3$ (Sigma Aldrich, 99%) at a molar ratio of 8:2:7:4. The precursors were blended in a high-energy ball milling machine at 300 rpm for 24 h; and the mixture was heat-treated at 750 °C for 1.5 h in a tube furnace with Ar flow. The as-prepared $Na_{1.5}VPO_{4.8}F_{0.7}$ was grinded into powder and mixed with poly-vinylidene fluoride (PVDF) binder and super P conductive additive at a weight ratio of 8:1:1 in $N$-methyl-2-pyrrolidone (NMP) solvent. The slurry was casted on carbon fiber cloth, because the conventional Cu or Al current collectors were reported unstable in CIBs[48]. The $Na_{1.5}VPO_{4.8}F_{0.7}$ electrodes were prepared with diameters of 3/8 inch and active mass loading of 2 mg cm$^{-2}$. Na ions were extracted from $Na_{1.5}VPO_{4.8}F_{0.7}$ electrodes by charging sodium ion half cells consisting of Na metal counter electrode and 1 M $NaPF_6$ EC/PC electrolyte up to 4.5 V vs. Na/Na$^+$. The desodiated NVPF electrodes were washed with flooded amount of dimethyl carbonate (DMC) solvent and dried in vacuum before using as working electrodes in CIBs.

The 1 M $Ca(PF_6)_2$ electrolyte was prepared by dissolving calcium salt in EC/PC (1/1, v/v) solvent in the glovebox and molecular sieves were also added to further remove $H_2O$ in electrolyte. The $Ca(PF_6)_2$ slat was synthesized according to the previous work[19]. A 25 mL acetonitrile suspension of 310 mg $CaCl_2$ was slowly dropped to a 25 mL acetonitrile solution containing 1.26 g $AgPF_6$. The mixture was magnetically stirred in dark for 24 h in the glovebox. After completion of reaction, the solution was centrifuged to obtain clear liquid, which was then dried in vacuum chamber, yielding a pure white powder of $Ca(PF_6)_2$.

**Materials characterization**. High resolution powder diffraction was performed at beamline 9B in the Pohang Accelerator Laboratory (PAL), Korea. The data were collected over 2θ between 10–103° with a step size of 0.01° and the wavelength of λ = 1.5226 Å. The XRD patterns were Rietveld refined using the FullProf program. Vanadium K-edge XANES spectra were obtained at beamline 7D in PAL using a double-crystal monochromator containing two sets of Si (111) crystals. The data were collected in transmission mode and normalized using the Athena program. $^{23}Na$ MRS NMR spectra for the discharged/charged electrode materials were performed using a Bruker Avance 400 MHz 9.4 T wide-bore spectrometer (4 mm probe). All the shifts were referenced to 0.1 M NaCl aqueous solution. For STEM characterization, cross-sectional (Cs-STEM) TEM specimens of the cycled electrode were prepared using focused ion beam (FIB) milling (FEI, Helio 650). The specimens were utilized for HAADF and ABF imaging under 60 keV using Cs-STEM (JEOL, JEM-ARM200F) with a point-to-point resolution of 0.08 nm.

**Electrochemical analysis**. CR2032 coin cells were assembled in glovebox using the desodiated NVPF as working electrode, the BP2000 activated carbon as counter electrode and Whatman glass fiber separator socked with 1 M $Ca(PF_6)_2$ EC/PC electrolyte. Ca-containing electrolyte served as the Ca resources in the coin cell, thus exceed amount of electrolyte was added, i.e., 400 uL per cell. The activated carbon voltage vs. Ca/Ca$^{2+}$ was estimated based on the previous literature[24] regarding the activated carbon electrode with an Ag/Ag$^+$ reference electrode and Fc/Fc$^+$ as internal reference dissolved in $Ca(PF_6)_2$ EC/PC electrolyte (Supplementary Fig. 2). The cells were galvanostatically discharged/charged between −1.0 and 1.5 V vs. activated carbon electrode at various current densities, and the specific capacities were determined based on the mass of active materials. GITT was carried out after a single discharge/charge cycle by applying a constant current density of 10 mA g$^{-1}$ for 30 min, followed by a relaxation potential measurement for 1 h, which protocol was repeated until the cell potential reached −1 or 1.5 V for discharging or charging, respectively. In situ EIS measurement was carried out on a Bio-Logic VSP-300 analyzer. Specifically, the coin cell was discharged/charged at a constant current density of 10 mA g$^{-1}$ for 25 min between −1 and 1.5 V and resting for 10 s, when EIS curves were periodically recorded over the frequency range of 10 kHz and 100 mHz at an amplitude of 5 mV.

**Synchrotron in situ XRD analysis**. In-situ XRD experiment was performed on a 5 A beamline at PAL with a wavelength of λ = 0.6885 Å, and the XRD data were collected approximately every 4 min as a set of circles on a Mar 345-image plate detector in transmission mode. Note that the coin cell was pre-cycled to active electrodes before sending to the in situ XRD experiment. The two-dimensional XRD images were converted to one-dimensional XRD curves and were recalculated to corresponding angles for λ = 1.543 Å, the wavelength of general X-ray tube sources with Cu Kα radiation. This process is to make it convenient for comparison of the in situ XRD results with these in literature.

**First principle calculations**. We used projector-augmented wave pseudopotentials[49,50] as implemented in Vienna Ab initio Simulation Package[51], and exchange-correlation energies were treated with generalized gradient approximation by Perdew–Burke–Ernzerhof[52]. In addition, a Hubbard $U$ parameter[53] of $U_{eff}$ = 4.0 eV was introduced to Vanadium in order to deal with the self-interaction error. The calculation of Ca diffusion was conducted in a large supercell containing 16 formula units of $Ca_xNa_{0.5}VPO_5F_{0.5}$ ($x = 0-0.5$) to prevent the interaction between migrating Ca ions in periodic unit cells. We note that $Ca_xNa_{0.5}VPO_5F_{0.5}$ ($x = 0-0.5$) structure was adapted instead of $Ca_xNa_{0.5}VPO_{4.8}F_{0.7}$

($x = 0-0.5$) to avoid the modeling of complicated O/F orderings. When $x = 0.5$, $Ca_xNa_{0.5}VPO_{4.8}F_{0.7}$ was fully calciated with full occupation of the capacity induced by Na extraction. For each diffusion path, seven intermediate images were placed to model the Ca diffusion pathway, and Nudged Elastic Band algorithm was used to obtain the activation barrier for Ca diffusion[54]. All calculations were performed with a cut-off energy of 500 eV and optimized until the remaining force in the unit cell converges within 0.05 eV Å$^{-1}$.

## Data availability
The data that support the plots within this paper and other findings of this study are available from the corresponding author on reasonable request.

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

## Acknowledgements

This work was supported by the National Research Foundation of Korea grant (Project No. 2018R1A2A1A05079249); the Supercomputing Center/Korea Institute of Science and Technology Information including technical support (KSC−2016-C3-0069) and code (IBS-R006-A2). This work was partially supported by a grant from the Research Committee of The Hong Kong Polytechnic University under project code 1-BE3M. This work was also supported by Shell International Exploration & Production, Inc.

## Author contributions

K.K. and Z.L.X conceived the original idea. Z.L.X conducted the experiment and analyzed the data. G.Y. carried out the simulation studies. J.W. and J.L. assisted on conducting synchrotron in situ XRD experiment, J.P., H.M., and S.Y.L. worked on fabricating graphite//NVPF CIB full cell, Y.J.K. and S.P.C. performed NMR and Cs-STEM experiments. K.K. revised the manuscript prepared by Z.L.X.

## Competing interests

The authors declare no competing interests.
