## [Peer Review File · Nature Communications]

Reviewer #1 only provided confidential comments to the editor

Reviewer #2 (Remarks to the Author):

This manuscript reports on a $\text{Na}_{1.5}\text{VPO}_{4.8}\text{F}_{0.7}$ derived material as a potential cathode candidate for Ca batteries. The authors have addressed the comments of the reviewers and this study demonstrates promising results in comparison with the state-of-the-art cathode materials for Ca batteries. Therefore, I recommend this paper for publication.

Authors' response to Reviewers' Comments:

The authors appreciate very much the invaluable comments offered by the reviewers and the editor.

Reviewer #1 (Remarks to the Author):

Reviewer #1 only provided confidential comments to the editor

Reply: We are grateful to the reviewer's efforts in reviewing our work.

Reviewer #2 (Remarks to the Author):

This manuscript reports on a $\text{Na}_{1.5}\text{VPO}_{4.8}\text{F}_{0.7}$ derived material as a potential cathode candidate for Ca batteries. The authors have addressed the comments of the reviewers and this study demonstrates promising results in comparison with the state-of-the-art cathode materials for Ca batteries. Therefore, I recommend this paper for publication.

Reply: We really appreciate the positive assessments from the reviewer to our Responses and the revised manuscript.